# Low Expression of UBE2Z, a Target Protein of miR-500a, Is Associated with Poor Prognosis in Triple-Negative Breast Cancer

**DOI:** 10.3390/ijms27010361

**Published:** 2025-12-29

**Authors:** Donghyun Kim, Song-Yi Choi

**Affiliations:** Department of Pathology, Chungnam National University School of Medicine, Daejeon 35015, Republic of Korea; dhkim11085@gmail.com

**Keywords:** triple-negative breast cancer, microRNAs, UBE2Z

## Abstract

Triple-negative breast cancer (TNBC) exhibits diverse histological and molecular characteristics. TNBC patients also have the poorest prognoses among those with various breast cancer subtypes, and no effective treatment strategy has been established for TNBC beyond non-specific chemotherapy. Recent studies have reported that the dysregulation of miRNAs is associated with tumor behavior, prognosis, and treatment responses in TNBC patients. Therefore, this study was conducted to identify miRNAs and key target proteins potentially associated with TNBC prognosis. Fresh-frozen tissue from relapsing and non-relapsing TNBC cases was examined for differentially expressed miRNAs using the Affymetrix GeneChip miRNA 4.0 array, while target genes and proteins were predicted using the miRwalk 2.0 database. The clinical significance of each differentially expressed miRNA was evaluated using the BreastMark database. Additional bioinformatics analyses were conducted to reveal associations with tumor-related signaling pathways; these analyses included protein–protein interaction network construction and Kyoto Encyclopedia of Genes and Genomes pathway annotation. Gene chip analysis identified three upregulated miRNAs (miR-500a, miR-501-3p, and miR-502-3p) and two downregulated miRNAs (miR-6798-5p and miR-7150) in patients with recurrence, and further bioinformatics analyses revealed that target proteins were significantly associated with cell cycle pathways. In addition, low expression of the miR-500a target protein UBE2Z was significantly associated with a poor prognosis. The expression levels of miR-500a and UBE2Z might be useful prognostic biomarkers in TNBC.

## 1. Introduction

Breast cancer is divided into several subtypes based on the protein expression profile. In clinical practice, immunohistochemical (IHC) staining for estrogen receptor (ER), progesterone receptor (PR), human epidermal growth factor receptor-2 (HER2), and the proliferation marker Ki-67 is used to broadly categorize breast tumors into four subtypes: luminal type A, luminal type B, the Her-2 type, and triple negative [1,2]. Triple-negative breast cancer (TNBC), which is defined by the undetectable expression of ER, PR, and HER2, accounts for about 15–20% of all breast cancers [3,4,5] and carries the worst prognoses among all subtypes as it exhibits rapid progression and carries higher risks of recurrence and distant metastasis [4,6]. In addition, TNBC exhibits highly variable histologic patterns, molecular characteristics, and clinical behavior [7,8,9]. Thus, unlike other subtypes, for which endocrine or targeted therapies have been established, TNBC treatment is limited to non-specific chemotherapy [10,11]. Furthermore, patients with TNBC exhibit variable responses to treatment, rendering it challenging to establish effective treatment strategies based on clinical studies and trials [12,13,14]. Therefore, it is necessary to elucidate the pathophysiology of TNBC in greater detail and identify biomarkers that can be used to predict the treatment response or serve as treatment targets.

microRNAs (miRNAs) are small, 21- or 22-nucleotide-long non-coding RNAs that negatively regulate the expression of target mRNAs. Dysregulation of miRNAs contributes to tumorigenesis, repressing the translation of target mRNAs and altering critical biological signaling pathways [15,16]. Recent studies have demonstrated associations between abnormal expression levels of specific miRNAs and tumor aggressiveness, prognosis, and response to treatment in TNBC patients [12,16,17]. While the majority of TNBCs are basal-like, characterized by the expression of proteins such as CK-5/6 or EGFR, and show poor prognoses, some are slow-growing and carry a more favorable prognosis. Increased expression levels of miR-21, miR-210, miR-454, and miR-27a/b are associated with poor survival among TNBC patients [16], suggesting that alterations in target proteins and signaling pathways underlie the diverse clinical presentations and treatment responses seen in TNBC. Thus, abnormal miRNA expression levels might serve as useful biomarkers for the prediction of clinical outcomes in TNBC patients.

In this study, we performed miRNA profiling to identify miRNAs that were differentially expressed between relapsing and non-relapsing patients after surgical treatment. We then analyzed the relationships between the predicted target proteins and clinicopathological factors.

## 2. Results

### 2.1. Analysis of Differentially Expressed miRNAs and Survival in TNBC

In fresh-frozen tissue samples from relapsing and non-relapsing TNBC patients, the expression levels of 2578 miRNAs were measured using a commercially available hybridization-based microarray. Five miRNAs showed significantly different expression levels between the two groups (Table 1). Of these miRNAs, the levels of miR-500a, miR-501-3p, and miR-502-3p were higher, whereas the levels of miR-6798-5p and miR-7150 were lower, in relapsing patients than in non-relapsing patients.

According to the open BreastMark database, which contains large-scale clinical data on breast cancer patients, elevated miR-500 expression is significantly associated with shorter DFS in breast cancer (all subtypes, n = 1563). In addition, patients (all subtypes and those with the basal-like type) with short OS tend to exhibit higher miR-500 levels (Appendix A).

### 2.2. Principal Component Analysis

A PCA-based scatterplot of the miRNA expression profiles (Figure 1) showed no significant clustering according to the TNBC subtype, although the miRNA expression patterns were distinct in each individual. Thus, patients with relapsing and non-relapsing TNBC were not distinguishable by their miRNA signatures.

### 2.3. Predicted Target Genes and Functional Analysis

The target genes of miRNAs that were differentially expressed between relapsing and non-relapsing patients were identified using the miRwalk 2.0 database (Appendix A). The three miRNAs that were overexpressed in relapsing patients targeted the following 13 genes (Appendix A): UGT2B10, IBA57, PSMG1, MTHFD2, ZNF460, SOD2, ELAVL2, NCAPG2, C8orf33, EFCAB11, NAA30, ZBTB43, and AGBL5. In addition, the two underexpressed miRNAs targeted the following six genes (Appendix A): BTG2, DPH2, TMEM109, ATG2A, KIAA1715, and CC2D1B. In total, these differentially expressed miRNAs were predicted to target 337 genes according to the DAVID database. The results of the gene functional analyses based on the Kyoto Encyclopedia of Genes and Genomes (KEGG) and the Biocarta pathway are shown in Table 2. The target genes of the five differentially expressed miRNAs demonstrated enrichment in the KEGG annotations “cell cycle” and “Rap1 signaling pathway”, whereas the Biocarta pathway analysis revealed associations with the tumor suppressor Arf, a major regulator of ribosomal biogenesis.

### 2.4. Analysis of Protein–Protein Interactions and Hub Proteins

A protein–protein network constructed from the predicted target proteins (Figure 2) revealed 18 hub proteins (Appendix A). To identify the proteins associated with prognosis in TNBC, we analyzed the correlations of the hub proteins with prognosis using the GOBO database. According to this database, UBE2Z and EPHB2 were associated with relapse-free survival in TNBC.

### 2.5. Associations Among UBE2Z Expression, TNBC Outcomes, and Clinical Features

The immunostaining of tissue samples revealed low (Allred scores ≤ 7) and high UBE2Z expression (Allred scores of 8) in 148 and 47 patients, respectively (Figure 3). Furthermore, we evaluated the correlations between UBE2Z expression and pathologic parameters (histologic grade, tumor size, and nodal metastasis) (Table 3). The frequency of lymph node metastasis was significantly higher (*p* = 0.008) in the low-expression group. However, the level of UBE2Z did not correlate with either the histologic grade or the tumor size. Furthermore, low UBE2Z expression was associated with significantly shorter OS compared to high UBE2Z expression (*p*-value = 0.037, Figure 4). Additionally, DFS tended to be shorter among patients with low UBE2Z expression (*p* = 0.064), although only slight statistical significance was observed.

## 3. Discussion

The clinical heterogeneity of TNBC hinders accurate prognosis, necessitating the identification of additional biomarkers for patient stratification [18]. Several recent studies have suggested that miRNA dysregulation influences the expression levels of various genes and molecular pathways associated with TNBC prognosis and chemotherapy resistance [19]. However, no miRNAs have been established as accurate and repeatable biomarkers for the stratification of patients with TNBC [12]. In the current study, we identified five miRNAs that were differentially expressed between relapsing and non-relapsing patients with TNBC. In addition, we found that UBE2Z, a downstream target protein of miR-500a-3p that is upregulated in TNBC, was significantly downregulated in relapsing cases. The expression levels of these miRNAs and UBE2Z could serve as prognostic markers and potential therapeutic targets in TNBC.

A comparison of the miRNA expression profiles between relapsing and non-relapsing patients revealed that miR-500a-3p, miR-501-3p, and miR-502-3p were upregulated and miR-6798-5p and miR-7150 were downregulated in relapsing patients. Several previous studies have reported associations between a poor prognosis and elevated levels of miR-21, miR-27a/b, miR-210, and miR-454, as well as between a favorable prognosis and reduced levels of miR-155 and miR-374a/b [20]. However, no previous investigation has demonstrated such associations with the miRNAs that were differentially expressed in our cohort. These disparities might be attributed to differences in analytic methods, as previous studies have primarily assessed differentially expressed miRNAs between tumor tissue and adjacent normal tissue, analyzing their associations with prognosis and subtype classification in TNBC [13,21,22]. Meanwhile, this is the first report comparing global miRNA profiles in tissue from relapsing and non-relapsing patients with TNBC (i.e., between patients with poor and more favorable prognoses).

The clustering of the five differentially expressed miRNAs (miR-500a, miR-501-3p, miR-502-3p, miR-6798-5p, and miR-7150) based on PCA did not enable clear separation into prognostic groups, suggesting that this differential expression profile is not a reliable signature for TNCB prognosis. In contrast, Cascione et al. found that the differential expression patterns of miR-16, miR-155, miR-125b, and miR-374a among tumor tissue, adjacent normal tissue, and lymph node metastasis tissue could be used to effectively predict OS and DFS [23]. In contrast, in the current study, only high miR-500a expression was strongly associated with relapse (poor prognosis), and high miR-500a expression was significantly associated with shorter DFS (*p* = 0.002) among a large cohort of breast cancer patients (n = 1563) from the BreastMark database. In this database, among patients with TNBC (n = 255), high miR-500a expression was similarly associated with shorter DFS, although only marginal statistical significance was observed (*p* = 0.069). The analysis of miR-500a expression in a larger number of TNBC tumor tissue samples is warranted to validate this association.

In contrast to miR-500a, there is little evidence of the contributions of the other four differentially expressed miRNAs to TNBC. Previously, Haldrup et al. [24] reported that the expression levels of miR-501-3p were elevated in the serum and tumor tissue of patients with prostate cancer. Hadavi et al. [25] found a correlation between miR-501-3p expression and the expression levels of PIK3CA and AKT1—key genes involved in the PI3K/AKT/mTOR pathway, which is dysregulated in TNBC.

Additionally, we analyzed the target genes and proteins of the differentially expressed miRNAs using bioinformatic tools, seeking to identify those associated with TNBC prognosis. We then utilized STRING to construct a functional protein association network and identify network hub genes. Among the 18 proteins identified as network hubs, UBE2Z exhibited a strong correlation with relapse-free survival according to the GOBO database. UBE2Z is a ubiquitin-conjugating enzyme with potential involvement in the regulation of apoptosis, among other critical cellular processes, but it has no known role in breast cancer. Thus, in this study, UBE2Z is identified as a potentially novel biomarker and therapeutic target. Shi et al. [26] reported that high UBE2Z expression in hepatocellular carcinoma was significantly correlated with advanced TNM stages and shorter OS and DFS. They also showed that UBE2Z regulates the JAK2/STAT3 and MAPK/ERK signaling pathways, both of which are implicated in cell cycle control, cell survival, and tumorigenesis. Contrary to previous reports, the cohort examined in the present study showed a favorable prognosis when UBE2Z expression was high. Nonetheless, this study has limitations. For example, it did not elucidate the mechanism behind the UBE2Z protein’s role in breast cancer, and the reasons for the conflicting findings observed in previous studies could not be clearly explained. However, we believe that the results are significant as the experiments were conducted on a large number of patient samples and demonstrated the possible clinical implications of UBE2Z expression. However, at present, research on the role of UBE2Z in cancer development and progression is very limited. Future research analyzing the role of UBE2Z in various types of cancer is warranted.

In conclusion, miRNA profiling allowed us to identify three upregulated miRNAs and two downregulated miRNAs in patients with TNBC with a poor prognosis (recurrence), as compared to patients with a more favorable prognosis. Among these, the levels of miR-500 were significantly higher in patients with recurrence. The ubiquitinating enzyme UBE2Z was identified as a target protein of several differentially expressed miRNAs, and its low expression was associated with shorter DFS and OS. Therefore, miR-500a and UBE2Z levels might serve as prognostic markers in TNBC.

## 4. Materials and Methods

### 4.1. Sample Selection

This study was approved by the Institutional Review Board of Chungnam National University Hospital (IRB no.: 2018-02-001). The medical records of patients with TNBC were retrospectively reviewed. For miRNA profiling, fresh-frozen tissue samples collected from the tumor site in ten relapsing and nine non-relapsing patients were selected, and patient groups were matched in terms of clinicodemographic factors such as age, histological grade, and stage. The biospecimens and data used for this study were provided by the biobank of Chungnam National University Hospital. To analyze the differences in the expression of predicted target genes at the protein level, formalin-fixed paraffin-embedded tissue from 195 patients with TNBC was retrieved from the Pathology Department’s archives together with reported pathologic characteristics, including the histological grade, tumor size, T stage, and N stage. These samples were then processed for immunohistochemistry.

### 4.2. microRNA Array Analysis

Total RNA was obtained from each of the 19 fresh-frozen tissue samples using Trizol reagent (Invitrogen, Waltham, MA, USA), following the manufacturer’s protocol. miRNA expression levels were then determined using the Affymetrix GeneChip miRNA 4.0 array (Affymetrix, Waltham, MA, USA), following the manufacturer’s instructions. The Affymetrix GeneChip miRNA 4.0 array contains 2578 mature human miRNA probes.

Raw data were extracted automatically into CEL files using the Affymetrix GeneChip^®^ Command Console^®^ (AGCC) software 4.1.2, followed by miRNA-level RMA+DABG-All analysis; the results were exported using Affymetrix^®^ Power Tools (APT) 1.18.0. Finally, the array data were filtered according to probe-annotated species. Relapsing and non-relapsing sample profiles were compared through independent-samples *t*-tests and fold changes, with the null hypothesis being that no difference would exist among the groups. Normalized signals were compared, and a *p*-value of 0.05 or less in the *t*-test was considered to indicate statistical significance. All statistical testing and the visualization of differentially expressed genes were conducted using the R statistical language, version 3.5.1 (https://www.r-project.org/). miRNAs with an absolute fold change value of 2 or more were considered significantly different.

### 4.3. Principal Component Analysis (PCA)

The expression profiles of the five miRNAs that were identified as differentially expressed were examined via PCA using the scikit-learn library in Python, version 1.6.0. The PCA results were visualized using a scatterplot in a reduced-dimensional space defined by the top three principal components.

### 4.4. In Silico Analysis

#### 4.4.1. Prognostic Value of Differentially Expressed miRNAs

We used the BreastMark database [27] to evaluate the associations between differentially expressed miRNAs and clinical outcomes, including disease-free survival (DFS) and overall survival (OS), considering all patients with breast cancer and specifically those with TNBC.

#### 4.4.2. Predicted Target Genes and Clinical Implications

The computational algorithms of MirWalk 2.0 (http://mirwalk.umm.uni-heidelberg.de, accessed on 25 March 2019) were used to predict the target genes of differentially expressed miRNAs. The Validated Target Module (VTM) search function was utilized to identify validated targets. Predicted target genes were then imported into DAVID (Database for Annotation, Visualization, and Integrated Discovery) (https://davidbioinformatics.nih.gov/) to extract biological features and pathway annotations associated with large gene lists. A *p*-value of <0.05 was considered to indicate a statistically significant association.

#### 4.4.3. Predicted Target Proteins and Implications

The STRING 11.0 database (https://string-db.org) was used to generate a functional interaction network for predicted target proteins and subsequently identify hubs connecting multiple proteins and pathways. To construct the protein–protein interaction (PPI) network, the required interaction score was set to a confidence value of 0.7, and the false discovery rate (FDR) was set to 5%. Cytoscape (version 3.7.1) was used to build the PPI network. The parameter settings were as follows: degree cutoff = 2, node score cutoff = 0.2, k-score = 2, maximum depth = 100 [18]. Genes with a degree > 10 were considered hub genes. The GOBO database (https://co.bmc.lu.se/gobo/, accessed on 27 March 2019) was used to analyze the associations among target protein expression and relapse-free survival.

### 4.5. Immunohistochemistry for UBE2Z

Using formalin-fixed paraffin-embedded (FFPE) sections from 195 TNBC patients, the expression levels of UBE2Z were estimated through IHC staining. In brief, all cases were reviewed, and 3 mm tissue microarrays (TMAs) were reconstructed. Two cores were obtained for each case, and 4 µm thick sections were prepared for IHC staining using a rabbit polyclonal anti-UBE2Z antibody (ab229877, Abcam, Cambridge, UK; diluted 1:150; control, placental tissue). UBE2Z immunoreactivity was scored using the semiquantitative Allred scoring method. The intensity score (0: negative, 1: weak, 2: moderate, 3: strong) and the proportion score (0: 0%, 1: <1%, 2: 1–10%, 3: <10–33%, 4: >33–66%, 5: >66%) were added to yield a total score ranging from 0 to 8. Cases with a score of 8 were classified as indicating “high expression of UBE2Z”, whereas those with a score lower than or equal to 7 were classified as having “low expression of UBE2Z”.

### 4.6. Statistical Analysis

Categorical variables were compared using the chi-squared or Fisher’s exact test. The threshold for statistical significance was set at *p* < 0.05. All statistical analyses were performed using IBM SPSS version 26 (IBM, Armonk, NY, USA), unless otherwise indicated.

## Figures and Tables

**Figure 1 ijms-27-00361-f001:**
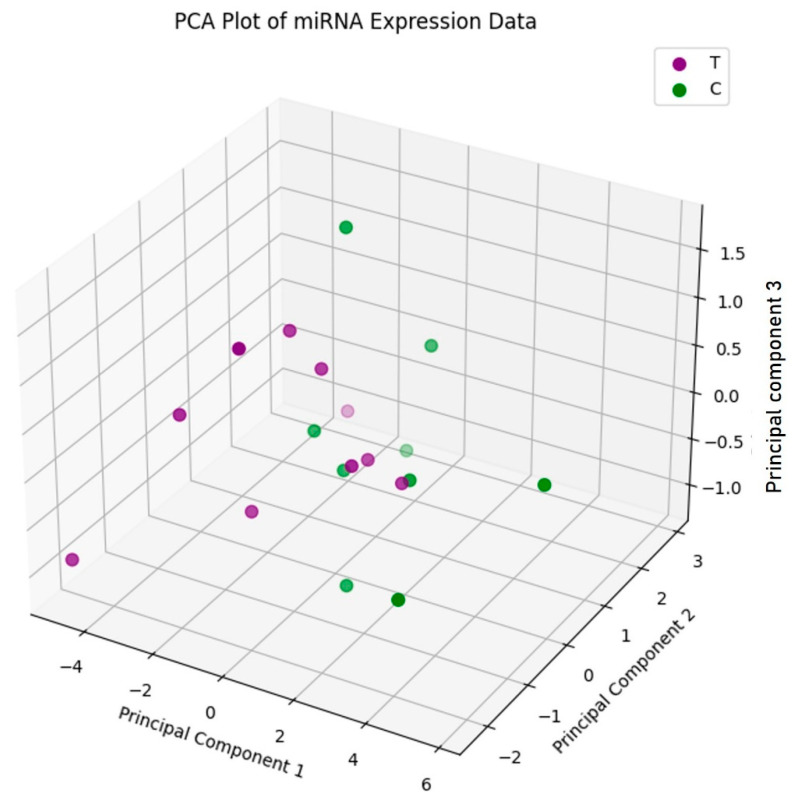
microRNA expression patterns in triple-negative breast cancer according to PCA analysis. T; TNBC with relapse, C; TNBC without relapse.

**Figure 2 ijms-27-00361-f002:**
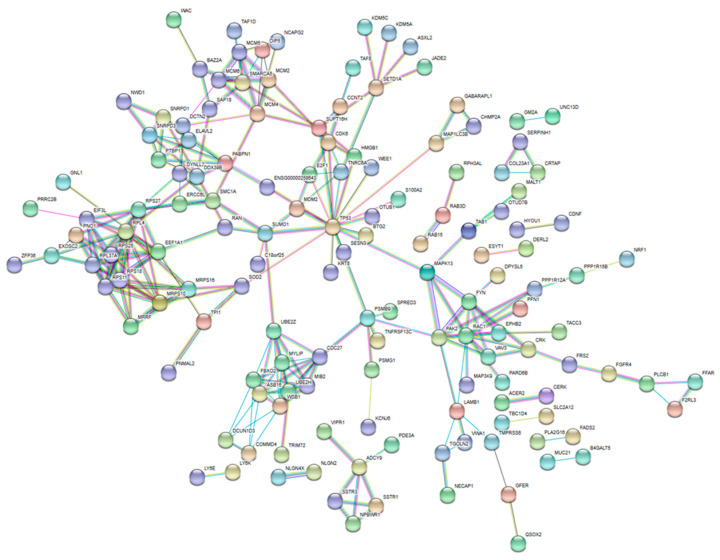
Cytoscape plot using STRING network data showing functional protein interrelationships.

**Figure 3 ijms-27-00361-f003:**
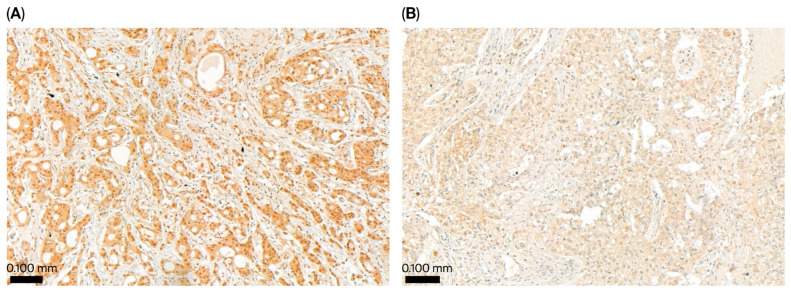
UBE2Z expression according to immunohistochemistry. High expression ((**A**) ×130) and low expression ((**B**) ×130) in tumor tissue from triple-negative breast cancer patients.

**Figure 4 ijms-27-00361-f004:**
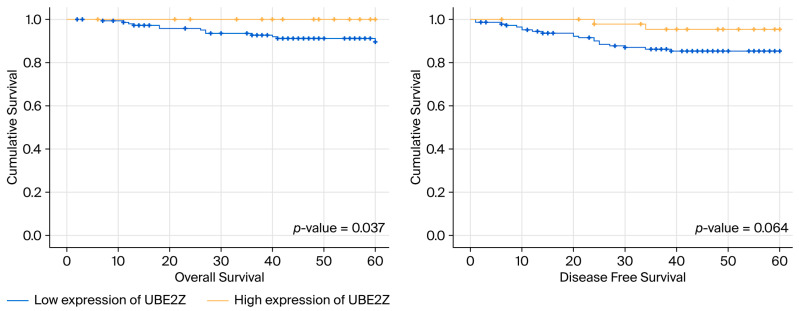
Kaplan–Meier survival curves according to UBE2Z expression in TNBC patients.

**Table 1 ijms-27-00361-t001:** Differentially expressed miRNAs in TNBC patients with relapse.

Expression in Comparison with Non-Relapse Patients	miRNA ID	Fold Change	*p*-Value
Overexpressed	hsa-miR-500a-3p	2.395126	0.039
	hsa-miR-501-3p	2.101952	0.037
	hsa-miR-502-3p	2.483641	0.047
Underexpressed	hsa-miR-6798-5p	−2.161684	0.050
	hsa-miR-7150	−2.336029	0.047

**Table 2 ijms-27-00361-t002:** Gene ontology analysis of predicted target genes of differentially expressed miRNAs.

Category	Term	Gene No.	*p*-Value
KEGG pathway	Cell cycle	9	1.2 × 10^−3^
	Rap1 signaling pathway	10	0.0010
	Neurotrophin signaling pathway	7	0.0170
	Glioma	5	0.0250
	Chronic myeloid leukemia	5	0.0350
Biocarta	Tumor suppressor Arf inhibits ribosomal biogenesis	4	0.0087
	Sumoylation by RanBP2 regulates transcriptional repression	3	0.0330

**Table 3 ijms-27-00361-t003:** Pathologic characteristics according to UBE2Z expression in TNBC.

UBE2Z Expression	Low (n = 148)	High (n = 47)	*p*-Value
Histologic grade			0.805
Grade 1	2 (1.4%)	0	
Grade 2	22 (14.9%)	7 (14.9%)	
Grade 3	124 (83.8%)	40 (85.1%)	
T stage			0.238
Early (T1)	62 (42.2%)	25 (53.2%)	
Advanced (T2–T4)	85 (57.8%)	22 (46.8%)	
LN metastasis			0.008
Negative (N0)	98 (67.1%)	41 (87.2%)	
Positive (N1–N3)	48 (32.9%)	6 (12.8%)	

## Data Availability

The original contributions presented in this study are included in the article and Appendix A. Further inquiries can be directed to the corresponding authors.

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
