# Peer review of "Low Expression of UBE2Z, a Target Protein of miR-500a, Is Associated with Poor Prognosis in Triple-Negative Breast Cancer"

_ijms, 2025, doi:10.3390/ijms27010361_

Round 1

Reviewer 1 Report

Comments and Suggestions for Authors

This manuscript explores miR-500a and its predicted target UBE2Z as potential prognostic biomarkers in TNBC. While the topic is interesting, the current version mainly depends on bioinformatic analysis and immunohistochemistry. The claims regarding a functional miR-500a-UBE2Z regulatory axis in TNBC lack experimental proof. Much more evidence is necessary before publication. Overall, the authors present an intriguing initial observation but do not provide cellular and molecular data to support their main conclusions. Therefore, the authors should perform mechanistic validation experiments to strengthen their findings.

  1. Lack of functional validation of the miR-500a-UBE2Z pathway in TNBC. Although the authors suggest UBE2Z is a direct downstream target of miR-500a, no cellular experiments confirm this regulation.
  2. No mechanistic insight into UBE2Z’s role in TNBC, and the data are too basic. While UBE2Z expression correlates with prognosis, there is no mechanistic data on its biological function in TNBC.

Author Response

Response to Reviewer 1 Comments

1. Summary

Thank you very much for taking the time to review this manuscript. Please find the detailed responses below.

2. Questions for General Evaluation

Reviewer’s Evaluation

Response and Revisions

Does the introduction provide sufficient background and include all relevant references?

Can be improved

Are all the cited references relevant to the research?

Can be improved

Is the research design appropriate?

Must be improved

Are the methods adequately described?

Must be improved

Are the results clearly presented?

Must be improved

Are the conclusions supported by the results?

Must be improved

3. Point-by-point response to Comments and Suggestions for Authors

Comments 1: Lack of functional validation of the miR-500a-UBE2Z pathway in TNBC. Although the authors suggest UBE2Z is a direct downstream target of miR-500a, no cellular experiments confirm this regulation.

Response 1: Thank you for pointing this out. I agree with your comments. While it would have been more accurate to demonstrate that UBE2Z is a direct target of miR-500a through cellular experiments, we chose a bioinformatics approach and utilized the validated target module (VTM) search function of the miRWalk database to identify validated targets. VTM provides experimentally validated miRNA interaction information. We have added the search algorithm we have used in the Methods. [ page 3 line 110-111, highlighted]

Comments 2: No mechanistic insight into UBE2Z’s role in TNBC, and the data are too basic. While UBE2Z expression correlates with prognosis, there is no mechanistic data on its biological function in TNBC.

Response 2:  I agree with your comments. This study lacks direct mechanistic experiments that validate gene-to-gene interactions. The authors aimed to identify prognostic factors in patients with triple-negative breast cancer who present with similar clinical presentations. We have additionally described the lack of mechanistic experiments using cells in the Discussion section. [page 8 line 263-269, highlighted]

4. Response to Comments on the Quality of English Language

Point 1: The English could be improved to more clearly express the research.

Response 1: We plan to utilize the author services for English editing to ensure clear communication.

  1. Additional clarifications

We plan to use author services to improve the clarity and comprehensibility of figures, tables, and graphical abstracts, based on general evaluation.

Reviewer 2 Report

Comments and Suggestions for Authors

The present study investigates the differences in expression of microRNAs (miRNAs) in triple-negative breast cancer (TNBC), and assesses UBE2Z as a potential prognostic biomarker. While the topic is of significance, the manuscript requires major revision due to methodological limitations, insufficient statistical rigor, and inconsistencies in interpretation.

L25–32: 'Result' should be replaced with 'Results'.

L39–41: Please note the repetition regarding TNBC prognosis.

L72–74: The Discovery cohort (n=19) was underpowered for microarray profiling.

L82–86: Please be advised that RIN values for RNA quality are not available.

L91–96: FDR-adjusted p-values not reported; significance questionable.

L119–129: Please note that the antibody catalog number and controls are missing.

L147–148: Please note that the FDR values in Table 1 are missing.

L157–170: It appears that pathway enrichment is overinterpreted in cases where there are small gene counts.

L173–178: The hub gene criteria in STRING have not been specified.

L184–193: The Allred score cutoff for UBE2Z is arbitrary and there is no ROC justification.

L188–193: Following a thorough review, it was determined that the DFS p-value is 0.064. This suggests that the conclusions may have been overstated.

L205–207: 'UZE2Z' should be 'UBE2Z'.

L208–209: It is important to note the logical inconsistencies in the direction of the UBE2Z–prognosis association.

L247–255: There is a contradiction with literature, where high UBE2Z correlates with worse survival.

Recommendations:

Please ensure that FDR-adjusted significance is included for the results of the microRNA tests.
Key miRNAs should be validated using qRT-PCR.

Multivariate Cox regression should be incorporated, incorporating clinicopathologic variables.

Please ensure that the PCA and KM plots are redrawn with full statistical annotation.

Please revise the discussion in order to resolve any contradictions with the published data.

Author Response

Response to Reviewer 2 Comments

1. Summary

Thank you very much for taking the time to review this manuscript. Please find the detailed responses below.

2. Questions for General Evaluation

Reviewer’s Evaluation

Response and Revisions

Does the introduction provide sufficient background and include all relevant references?

Can be improved

Is the research design appropriate?

Can be improved

Are the methods adequately described?

Must be improved

Are the results clearly presented?

Can be improved

Are the conclusions supported by the results?

Must be improved

3. Point-by-point response to Comments and Suggestions for Authors

Comments 1:  'Result' should be replaced with 'Results'.

Response 1:  Corrected [ line 141, highlighted]

Comments 2: Please note the repetition regarding TNBC prognosis.

Response 2: Thank you for pointing this out. Edited to remove redundant content. [Deleted content: complicating prognosis and treatment] [ line 44]

Comments 3: The Discovery cohort (n=19) was underpowered for microarray profiling.

Response 3: We agree that the sample size was insufficient. A larger sample size would have yielded more reliable results. Please note that we had limited fresh-frozen tissue available.

Comments 4: Please be advised that RIN values for RNA quality are not available.

Response 4: MiRNAs are known to be stable even during mRNA degradation due to their short nucleotide sequences. To verify data quality, we checked density plots and control probe line plots. [If necessary, additional information will be provided in a supplementary figure.]

Comments 5: FDR-adjusted p-values not reported; significance questionable

Response 5: Thak you for pointing this out. I agree with your comment. We obtained p-values by performing t-tests using the non-relapsed group as a control group. While using FDR correction can reduce false positives, we applied p-values to account for the potential for missing candidate targets. The analysis method was incorrectly described in the methods, so we corrected it. [line 92-93, highlighted]

Comments 6: Please note that the antibody catalog number and controls are missing.

Response 6: Catalog numbers and positive control tissue were additionally described. [line 130-131, highlighted]

Comments 7: Please note that the FDR values in Table 1 are missing.

Response 7: As explained in comment 5, the p-value of the t-test was applied to evaluate significant expression differences in this study. [line 92-93, highlighted]

Comments 8:  It appears that pathway enrichment is overinterpreted in cases where there are small gene counts.

Response 8: We analyzed 337 genes presented as queries, including those individually described in the text.

Comments 9:  The hub gene criteria in STRING have not been specified.

Response 9: Thank you for pointing this out. The criteria for protein network analysis were additionally described in the method. [line 118-122, highlighted]

Comments 10:  The Allred score cutoff for UBE2Z is arbitrary and there is no ROC justification.

Response 10: Thank you for pointing this out. On the immunohistochemical staining with UBE2Z, a diffuse strong positive pattern corresponding to Allred score of 8 was observed in normal mammary epithelium, so expression 7 or less Allred score was considered as abnormal low expression.

Comments 11:   Following a thorough review, it was determined that the DFS p-value is 0.064. This suggests that the conclusions may have been overstated.

Response 11: I agree with your comment. The results section states, "Patients with low UBE2Z expression tended to have shorter disease-free survival (DFS) (p = 0.064)”. In addition, we edited the phrase to avoid overstating the statistical significance. [line 200, highlighted]

Comments 12:   'UZE2Z' should be 'UBE2Z'.

Response 12: Corrected [line 215,216, highlighted]

Comments 13:    It is important to note the logical inconsistencies in the direction of the UBE2Z–prognosis association.

Response 13: Thank you for pointing this out. The sentence has been revised for consistency of logical direction. [line 214-216, highlighted]

Comments 14:    There is a contradiction with literature, where high UBE2Z correlates with worse survival.

Response 14: Thank you for pointing this out. Only one study, targeting hepatocellular carcinoma, has reported an association between UBE2Z expression and prognosis. While the previous results are inconsistent with our results, the two studies investigated different cancer types, making reproducibility difficult to assess. Future research data collected via various cancer types will likely elucidate the role of UBE2Z. Furthermore, these limitations of our study have been added in the Discussion. [line 263-269, highlighted]

4. Response to Comments on the Quality of English Language

Point 1: The English could be improved to more clearly express the research.

Response 1: We plan to utilize the author services for English editing to ensure clear communication.

5. Additional clarifications

We plan to use author services to improve the clarity and comprehensibility of figures, tables, and graphical abstracts, based on general evaluation.

Round 2

Reviewer 1 Report

Comments and Suggestions for Authors

During this revision, the authors examine the prognostic importance of miR-500a and its potential target UBE2Z in TNBC using publicly available datasets and bioinformatics tools. While the correlation analyses are useful and the clinical relevance is clear, the study is limited by a lack of functional validation, especially concerning the miR-500a-UBE2Z regulatory axis and UBE2Z's mechanistic role in TNBC progression. However, the authors have recognized these limitations and addressed them in the revised manuscript, which is appreciated. At this stage, the authors provide initial insights that could guide future research but do not present a detailed mechanistic understanding. Overall, I recommend that the authors conduct follow-up studies to directly examine UBE2Z's functional role in TNBC using experimental models in future work, as such data would significantly enhance the scientific value of this research.

Reviewer 2 Report

Comments and Suggestions for Authors

All the required modifications were made. The article is suitable for publication.